# The Bifurcated σ-Hole···σ-Hole Stacking Interactions

**DOI:** 10.3390/molecules27041252

**Published:** 2022-02-13

**Authors:** Yu Zhang, Weizhou Wang

**Affiliations:** College of Chemistry and Chemical Engineering, and Henan Key Laboratory of Function-Oriented Porous Materials, Luoyang Normal University, Luoyang 471934, China; yzhpaper@yahoo.com

**Keywords:** bifurcated σ-hole···σ-hole stacking interaction, organosulfur molecules, Cambridge structural database, quantum chemical calculation

## Abstract

The bifurcated σ-hole···σ-hole stacking interactions between organosulfur molecules, which are key components of organic optical and electronic materials, were investigated by using a combined method of the Cambridge Structural Database search and quantum chemical calculation. Due to the geometric constraints, the binding energy of one bifurcated σ-hole···σ-hole stacking interaction is in general smaller than the sum of the binding energies of two free monofurcated σ-hole···σ-hole stacking interactions. The bifurcated σ-hole···σ-hole stacking interactions are still of the dispersion-dominated noncovalent interactions. However, in contrast to the linear monofurcated σ-hole···σ-hole stacking interaction, the contribution of the electrostatic energy to the total attractive interaction energy increases significantly and the dispersion component of the total attractive interaction energy decreases significantly for the bifurcated σ-hole···σ-hole stacking interaction. Another important finding of this study is that the low-cost spin-component scaled zeroth-order symmetry-adapted perturbation theory performs perfectly in the study of the bifurcated σ-hole···σ-hole stacking interactions. This work will provide valuable information for the design and synthesis of novel organic optical and electronic materials.

## 1. Introduction

Since the creation of the term “σ-hole” by Clark et al. [1,2], the study of hole interactions has been the focus of recent attention in the field of noncovalent interactions [3]. The σ-hole bond was generally defined as a net attractive interaction between one electrophilic σ-hole and the other nucleophilic region [4,5]. Evidently, the net attractive interaction between an electrophilic region and an electrophilic region or between a nucleophilic region and a nucleophilic region cannot be classified as the σ-hole bond. The electrostatic potential associated with a σ-hole can be positive or negative [6]. We termed the net attractive interaction between an electrophilic region associated with one σ-hole and an electrophilic region associated with the other σ-hole or between a nucleophilic region associated with one σ-hole and a nucleophilic region associated with the other σ-hole as the σ-hole···σ-hole stacking interaction [7,8]. The σ-hole···σ-hole stacking interaction is quite similar to the π···π stacking interaction. Figure 1 is a schematic diagram of the sandwich σ-hole···σ-hole stacking interaction versus sandwich π···π stacking interaction, and parallel-displaced σ-hole···σ-hole stacking interaction versus parallel-displaced π···π stacking interaction. Here, only the interactions between two positive σ-holes are shown. The case is similar for the interactions between two negative σ-holes; for example, in a recent study, Frontera and coworkers reported the σ-hole···σ-hole stacking interaction between two I_3_^–^ anions in the crystal structure of a hetero-trinuclear cobalt(III)/potassium complex [9]. Very recently the term of σ-hole···σ-hole stacking interaction has also been discussed in other papers [10,11]. The word “stacking” in σ-hole···σ-hole stacking interaction has three-fold implications: (i) the net attractive interaction occurs between two nucleophilic regions or between two electrophilic regions; (ii) the two σ bonds associated with two σ-holes are linear or nearly linear; (iii) the σ-hole···σ-hole stacking interaction is dispersion-dominated as it is for the π···π stacking interaction [7,8].

The σ-hole···σ-hole stacking interactions and π···π stacking interactions are similar in many aspects but do have some differences. Different from the π···π stacking interactions, there are a large number of bifurcated σ-hole···σ-hole stacking interactions found in the crystal structures (Figure 2). A monovalent halogen atom has only one σ-hole, so the σ-hole···σ-hole stacking interaction between two monovalent halogen atoms is monofurcated. A divalent chalcogen atom can bear two σ-holes and, as a result, the intermolecular or intramolecular bifurcated σ-hole···σ-hole stacking interaction can be formed by the two σ bonds on the divalent chalcogen atom with the other two σ bonds (Figure 2). Figure 2b is a typical example of the intramolecular bifurcated σ-hole···σ-hole stacking interactions in the crystal structure of 5,16-diphenyltetrathia[22]annulene which has potential applications in organic field-effect transistors [12]. EJIGOU is the reference code for the crystal structure of 5,16-diphenyltetrathia[22]annulene in the Cambridge Structural Database (CSD) [13]. The tetrathia[22]annulene has a nearly planar structure, which indicates that the intramolecular bifurcated σ-hole···σ-hole stacking interactions in tetrathia[22]annulene are not repulsive, and are helpful for the stabilization of tetrathia[22]annulene. Considering that it is more convenient to calculate the binding strength of the intermolecular bifurcated σ-hole···σ-hole stacking interaction, we did not investigate the intramolecular bifurcated σ-hole···σ-hole stacking interaction in this work.

It is well-known that organosulfur compounds have important applications in medicinal, agricultural, and materials chemistry [14]; especially in recent years the thiophene-based organic molecules and polymers have been found to play key roles in the development of novel organic optical and electronic materials [15]. A thorough understanding of the noncovalent interactions between organosulfur molecules is necessary for the origin and development of the organic optical and electronic materials [15]. Therefore, it is important to study the bifurcated σ-hole···σ-hole stacking interactions between the C–S σ single bonds (Figure 2).

## 2. Materials and Methods

To carry out a CSD search for the bifurcated σ-hole···σ-hole stacking interactions between the C–S σ single bonds, the difficulty lies in how to define the ranges of the angles ANG1, ANG2, ANG3 and ANG4 in Figure 2a [16,17]. The angular preferences of halogen bonds with the type of C–X···O–Y in biomolecular complexes have been explored by Ho and coworkers [16]. Histogram distribution of the X···O–Y angles shows that a very small number of halogen bonds were observed with the X···O–Y angles larger than 160° [16]. The hint for the C–S···S–C contacts is that it is reasonable to assume that the C–S···S–C σ-hole···σ-hole stacking interactions not the C–S···S–C chalcogen bonds will exist in the angular range of 160–180° for the S···S–C angles. Note that the C–S···S–C chalcogen bond is a sister noncovalent bond of the C–X···O–Y halogen bond [18]. Hence in this study we selected 160–180° as the ranges of the angles ANG1, ANG2, ANG3 and ANG4. At the same time, the S···S distance was set to be less than the sum of the van der Waals radii of two S atoms. Using these criteria, the CSD search retrieved 17 individual crystal structures containing the bifurcated σ-hole···σ-hole stacking interactions. The CSD entry codes of these crystal structures are CADFIZ, ELASUG, GABDAP, HOWGEG, KASDIS, MECJUZ, MUFFID, NEFPUJ, TONPES, TRITAN03, UPICAY, UVABOJ, VIDWUC, WEPGON, XEFFUL, XEGNAA and XEGNEE. In this work, we investigated in detail the bifurcated σ-hole···σ-hole stacking interactions in the crystal structures of UPICAY, WEPGON, XEGNAA and XEGNEE. UPICAY denotes the crystal structure of *N*-octanoyldithieno[3,2-*b*:2′,3′-*d*]pyrrole [19]; WEPGON represents the crystal structure of 4-(1,3-dithiol-2-ylidene)-4H-cyclopenta(2,1-*b*;3,4-*b*′)dithiophene [20]; XEGNAA denotes the crystal structure of 4,5-bis(4-methylphenyl)-4,5-dihydrothieno[3,2-*b*]thieno[2″,3″:4′,5′]pyrrolo[2′,3′:4,5]thieno[2,3-*d*]pyrrole and XEGNEE represents the crystal structure of 4,5-bis(4-*n*-butylphenyl)-4,5-dihydrothieno[3,2-*b*]thieno[2″,3″:4′,5′]pyrrolo[2′,3′:4,5]thieno[2,3-*d*]pyrrole [21]. The selections of UPICAY, WEPGON, XEGNAA and XEGNEE are random and the results for the bifurcated σ-hole···σ-hole stacking interactions in the other crystal structures should be the same.

Accurate calculations of the weakly bound complexes are still great challenges in the field of noncovalent interactions. It is very common that many computational methods which perform excellently on the testing weakly bound complexes fail to describe the noncovalent interactions in a new system. Thus, it is necessary to analyze the reliability and feasibility of the computational methods before the research has been conducted. Based on our previous studies on a series of π-stacked complexes [22,23], the dispersion-corrected density functional PBE0-D3 in conjunction with the basis set def2-TZVPP and Becke–Johnson damping function (PBE0-D3(BJ)/def2-TZVPP) will be employed for the study of the bifurcated σ-hole···σ-hole stacking interactions in the crystal structures of UPICAY, WEPGON, XEGNAA and XEGNEE [24,25,26,27].

The small 2-fluorothiophene dimer was selected as a model to explore the reliability of the PBE0-D3(BJ)/def2-TZVPP calculations. As shown in Figure 3, we utilized two F atoms to substitute two H atoms on the 2-positions of the thiophene dimer in order to avoid the formation of the C–H···S hydrogen bond between two thiophene molecules. At the PBE0-D3(BJ)/def2-TZVPP level of theory, the C1–S1 and C2–S2 σ bonds are almost in a straight line, which indicates the existence of a typical C1–S1···S2–C2 σ-hole···σ-hole stacking interaction not a C1–S1···S2–C2 chalcogen bond. The PBE0-D3(BJ)/def2-TZVPP interaction energy in Figure 3 has been corrected for the basis set superposition error by applying the Boys–Bernardi’s counterpoise (CP) procedure [28]. Using the PBE0-D3(BJ)/def2-TZVPP optimized geometries, the gold-standard interaction energy of the 2-fluorothiophene dimer was determined at the complete basis set limit coupled-cluster [CCSD(T)/CBS] theory level. The computational procedure of the CCSD(T)/CBS method is completely the same as those employed by Sherrill’s group and Hobza’s group [29,30]. It can be clearly seen from Figure 3 that the difference between PBE0-D3(BJ)/def2-TZVPP interaction energy and gold-standard CCSD(T)/CBS interaction energy is only 0.10 kcal/mol, while the PBE0-D3(BJ)/def2-TZVPP calculation is much cheaper with respect to the computational costs than the CCSD(T)/CBS calculation. This means that the PBE0-D3(BJ)/def2-TZVPP calculation has a very high accuracy/cost ratio, and is reliable and feasible for the study of large complexes bound by the σ-hole···σ-hole stacking interactions.

Throughout the paper, we used UPICAY-M, WEPGON-M, XEGNAA-M and XEGNEE-M to represent the corresponding monomers in the crystal structures (Figure 4), and UPICAY-D, WEPGON-D, XEGNAA-D and XEGNEE-D to represent the corresponding dimers in the crystal structures (Figure 5). The geometries of these monomers and dimers were all extracted from the crystal structures and without further optimizations. The molecular electrostatic potentials of the monomers were calculated at the PBE0-D3(BJ)/def2-TZVPP theory level. The CP-corrected interaction energies of the dimers were calculated at the PBE0-D3(BJ)/def2-TZVPP theory level, which has been proven accurate and efficient for the study of the large complexes bound by σ-hole···σ-hole stacking interactions. For comparison, the CP-corrected HF/def2-TZVPP interaction energies of the dimers were also calculated.

The energy decomposition analysis of the bifurcated σ-hole···σ-hole stacking interaction was performed by using the spin-component scaled zeroth-order symmetry-adapted perturbation theory (SCS-SAPT0) with the basis set aug-cc-pVDZ [31,32,33,34]. Sherrill and coworkers developed various SAPT methods for the energy component analyses of noncovalent interactions [31,32,33]. The computations of the high-order SAPT methods are very expensive and cannot be employed for the study of the large systems. The low-order SAPT0 method neglects the intra-monomer electron correlation and can treat the systems with hundreds of atoms. It has been reported that the combination of the SCS-SAPT0 method with the basis set aug-cc-pVDZ has a very high accuracy in analyzing the noncovalent interactions in the large π-stacked complexes [34]. The bonding characteristic of the bifurcated σ-hole···σ-hole stacking interaction was analyzed with Bader’s “atoms in molecules” (AIM) theory [35]. The AIM theory defines chemical bond and structure of a chemical system based on the topological analysis of the electron density and its Laplacian. It has gradually become a very valuable tool for addressing possible questions regarding the noncovalent interactions [35].

The molecular electrostatic potentials and interaction energies were calculated with the GAUSSIAN 09 program package [36]; SCS-SAPT0/aug-cc-pVDZ calculations were performed by utilizing the PSI4 suite of programs [33]; AIM analyses were carried out by using the AIM2000 software [37].

## 3. Results and Discussion

Figure 4 shows the molecular electrostatic potential maps of the monomers UPICAY-M, WEPGON-M, XEGNAA-M and XEGNEE-M. As recommended by both Bader’s group and Politzer’s group, the electrostatic potentials of these monomers are computed on their molecular surfaces defined by 0.001 au contours of the molecular electronic densities [38,39]. As can clearly be seen in Figure 4, there are two positive σ-holes on each labeled S atom, along the extensions of two C–S σ bonds. The existence of two σ-holes on one S atom is a basis for the formation of the bifurcated σ-hole···σ-hole stacking interaction. Figure 5 illustrates the structures of the dimers UPICAY-D, WEPGON-D, XEGNAA-D and XEGNEE-D bound by the bifurcated σ-hole···σ-hole stacking interactions. The corresponding S···S interatomic distances, C–S···S angles and interaction energies of these dimers are summarized in Table 1.

The two interacting conjugated backbones in each of the dimers in Figure 4 are in the same plane, or nearly so. The C–S···S angles in Table 1 are all close to 180°. At the same time, Figure 4 and Figure 5 jointly show that one positive σ-hole on S atom along the extension of one C–S σ bond points to the other positive σ-hole on S atom along the extension of the other C–S σ bond, not the electron-rich lone pair region on S atom. All these results show that the bifurcated σ-hole···σ-hole stacking interactions not the C–S···S–C chalcogen bonds are formed in the dimers UPICAY-D, WEPGON-D, XEGNAA-D and XEGNEE-D. The sum of the van der Waals radii of two S atoms is 3.60 Å [40]. It is noticed in Table 1 that the S2···S5 interatomic distance in XEGNAA-D is only 3.161 Å, which is 12% shorter than the sum of the van der Waals radii of two S atoms. The S···S interatomic distance will determine whether the monofurcated σ-hole(S)···σ-hole(S) stacking interaction is attractive or repulsive. It is difficult to separate the binding energy of the monofurcated σ-hole(S2)···σ-hole(S5) stacking interaction from the total binding energy of the XEGNAA-D dimer, but we can get some hints from the PBE0-D3(BJ)/def2-TZVPP calculations of the 2-fluorothiophene dimer. At the S···S interatomic distances of 3.1, 3.2, 3.3, 3.4, 3.5 and 3.6 Å, the PBE0-D3(BJ)/def2-TZVPP interaction energies of the 2-fluorothiophene dimer are +0.16, −0.48, −0.90, −1.17, −1.33 and −1.41 kcal/mol, respectively, which supports above observation. The interaction energies of the dimers UPICAY-D, WEPGON-D and XEGNEE-D are −3.60, −3.16 and −3.40 kcal/mol, respectively. Each of the dimers, UPICAY-D, WEPGON-D and XEGNEE-D, has three monofurcated σ-hole···σ-hole stacking interactions. This means that the absolute value of the interaction energy of one monofurcated σ-hole···σ-hole stacking interaction is larger than 1.00 kcal/mol. The interaction energies of the dimer XEGNAA-D are −4.73 kcal/mol. Normally the absolute value of the interaction energy of XEGNAA-D should be larger than 5.00 kcal/mol because it has five monofurcated σ-hole···σ-hole stacking interactions. Evidently, it is the much weaker monofurcated σ-hole(S2)···σ-hole(S5) stacking interaction that results in much smaller binding energy of the XEGNAA-D dimer. In fact, the monofurcated noncovalent interactions in all the bifurcated noncovalent interactions are not in their most stable structures. Due to the geometric constraints, the binding energy of one bifurcated noncovalent interaction is in general smaller than the sum of the binding energies of two free monofurcated noncovalent interactions.

Table 2 summarizes the results of energy decomposition analyses for the total interaction energies of the dimers UPICAY-D, WEPGON-D, XEGNAA-D and XEGNEE-D at the SCS-SAPT0/aug-cc-pVDZ level of theory. Here the total interaction energy of the dimer is decomposed into four energy components: electrostatic energy (*E*_elst_), exchange repulsion energy (*E*_exch_), induction energy (*E*_ind_), and dispersion energy (*E*_scs-disp_). Additionally listed in Table 2 are the total interaction energies at the PBE0-D3(BJ)/def2-TZVPP and SCS-SAPT0/aug-cc-pVDZ levels of theory, respectively. The SCS-SAPT0/aug-cc-pVDZ total interaction energies are very close to the corresponding PBE0-D3(BJ)/def2-TZVPP total interaction energies, and the largest deviation is only 0.26 kcal/mol which is 7.6% of the PBE0-D3(BJ)/def2-TZVPP total interaction energy of XEGNEE-D. The difference between PBE0-D3(BJ)/def2-TZVPP total interaction energy and HF/def2-TZVPP total interaction energy [*E*_disp_(2)] was also calculated to approximately assess the reliability of the SCS-SAPT0/aug-cc-pVDZ dispersion energy. The values of *E*_disp_(2) for the dimers UPICAY-D, WEPGON-D, XEGNAA-D and XEGNEE-D are −7.40, −7.52, −12.14 and −9.00 kcal/mol, respectively. Correspondingly the SCS-SAPT0/aug-cc-pVDZ dispersion energies for the dimers UPICAY-D, WEPGON-D, XEGNAA-D and XEGNEE-D are −7.50, −7.73, −12.49 and −9.43 kcal/mol, respectively. Clearly the dispersion energies calculated by two different methods are quite close to each other. All these results show that the SCS-SAPT0/aug-cc-pVDZ calculations are reliable for the study of the bifurcated σ-hole···σ-hole stacking interactions in the dimers UPICAY-D, WEPGON-D, XEGNAA-D and XEGNEE-D.

As shown in Table 2, the dispersion energies account for about 60% of the total attractive interaction energies of the four dimers, which show that the bifurcated σ-hole···σ-hole stacking interactions are still of the dispersion-dominated type. The contributions of the induction energies to the total attractive interaction energies are the smallest ones among the attractive energy components in the four dimers. The electrostatic energies contribute about 30% of the total attractive interaction energies and play secondary roles in stabilizing the bifurcated σ-hole···σ-hole stacking interactions. The contributions of the electrostatic energies to the total attractive interaction energies are in the range of 10–15% for the monofurcated σ-hole(Br)···σ-hole(Br) stacking interactions [8]. In contrast, the contributions of the electrostatic energies in this study increase to about 30% of the total attractive interaction energies. It is noticed that the Br···Br interatomic distances in the monofurcated σ-hole(Br)···σ-hole(Br) stacking interactions are much larger than the S···S interatomic distances in the bifurcated σ-hole···σ-hole stacking interactions. A previous study has shown that the charge penetration effect plays an important role in the understanding of the electrostatic contributions, and the charge penetration effects become larger at shorter distances [41]. Thus, the larger contributions of the electrostatic energies in the bifurcated σ-hole···σ-hole stacking interactions can be explained by the larger charge penetration effects.

The AIM analyses have been carried out to further uncover the bonding characteristic of the bifurcated σ-hole···σ-hole stacking interaction. Figure 6 shows the bond paths and bond critical points of the dimers UPICAY-D, WEPGON-D, XEGNAA-D and XEGNEE-D. The structures and atomic labels in Figure 6 are the same as the corresponding ones in Figure 5. Table 3 lists the electron densities, electron density Laplacians, eigenvalues of the Hessian matrix and ellipticities at the bond critical points of the S···S contacts in the four dimers.

It is clear in Table 3 that the values for *ρ*_b_ are all low and the values for ▽^2^*ρ*_b_ are all larger than zero at the bond critical points of the S···S contacts in the four dimers, which exhibits the characteristics of the existence of noncovalent interactions. The bond paths in Figure 6 indicate the two S atoms are bound together. However, it is also noticed in Figure 6 that the bond paths between two S atoms are all curved away from the S···S internuclear axes. The curved bond path is a significance of weak noncovalent interaction; it is clearly consistent with the result from the interaction energy calculation. The electron density Laplacian (▽^2^*ρ*) is the sum of three eigenvalues of the Hessian matrix (*λ*_1_, *λ*_2_, *λ*_3_) and ellipticity (*ε*) can be calculated as *λ*_1_/*λ*_2_−1. Besides the π-character of the bonding, the ellipticity is also a measure of the structural stability [42]. A previous study showed that the noncovalent bond with larger ellipticity will be much weaker [43]. In Table 3, the S2···S5 bonding ellipticity in XEGNAA-D is the largest one, which echoes the above result that the monofurcated σ-hole(S2)···σ-hole(S5) stacking interaction is slightly attractive or even repulsive.

Figure 7 shows the correlation of the S···S interatomic distance with the electron density or electron density Laplacian. Both the Pearson correlation coefficients and the adjusted correlation coefficients are close to one, which shows that the correlation of the S···S interatomic distance with the electron density or electron density Laplacian is very strong. However, as discussed above, there is no linear correlation between S···S interatomic distance and interaction energy of the σ-hole···σ-hole stacking interaction. This is different from previous studies in which it was claimed that the interatomic distances, electron densities at the bond critical points and interaction energies of some noncovalent bonds have linear correlations with each other [44,45]. The main reason is that the monofurcated σ-hole···σ-hole stacking interactions in the bifurcated σ-hole···σ-hole stacking interaction are subject to some geometrical restrictions and not in their optimal structures.

## 4. Conclusions

In this work, the bifurcated σ-hole···σ-hole stacking interactions between organosulfur molecules were studied in detail with a combined method of the Cambridge Structural Database search and quantum chemical calculation. The organosulfur molecules selected in this work are important building blocks for organic optical and electronic materials. It was found that the binding energy of one bifurcated σ-hole···σ-hole stacking interaction is in general smaller than the sum of the binding energies of two free monofurcated σ-hole···σ-hole stacking interactions. The main reason is that the monofurcated σ-hole···σ-hole stacking interactions in the bifurcated σ-hole···σ-hole stacking interaction are subject to some geometrical restrictions and not in their optimal structures. The results of energy-component analyses for the total interaction energies of the dimers UPICAY-D, WEPGON-D, XEGNAA-D and XEGNEE-D show that the bifurcated σ-hole···σ-hole stacking interactions are still dispersion dominated, although the contributions of the electrostatic energies to the total attractive interaction energies increase significantly and dispersion energy contributions decrease significantly in comparison with the linear monofurcated σ-hole···σ-hole stacking interactions.

The calculations of the 2-fluorothiophene dimer which is taken as a model show that the PBE0-D3(BJ)/def2-TZVPP performs perfectly for the study of the complex bound by the σ-hole···σ-hole stacking interaction, and the accuracy of the PBE0-D3(BJ)/def2-TZVPP method is comparable to the accuracy of the “gold-standard” CCSD(T)/CBS method. Another important finding of this study is that the PBE0-D3(BJ)/def2-TZVPP interaction energies for the dimers UPICAY-D, WEPGON-D, XEGNAA-D and XEGNEE-D are almost the same as the corresponding SCS-SAPT0/aug-cc-pVDZ interaction energies. The calculations of the dispersion energies prove the reliability of the energy components calculated by the SCS-SAPT0/aug-cc-pVDZ method. The SCS-SAPT0/aug-cc-pVDZ calculation is very cheap and it can be used for the study of very large weakly bound complexes.

Besides the bifurcated σ-hole···σ-hole stacking interactions between C–S σ bonds, bifurcated σ-hole···σ-hole stacking interactions may exist between other σ bonds. Such kinds of bifurcated σ-hole···σ-hole stacking interactions are under investigation in our laboratory.

## Figures and Tables

**Figure 1 molecules-27-01252-f001:**
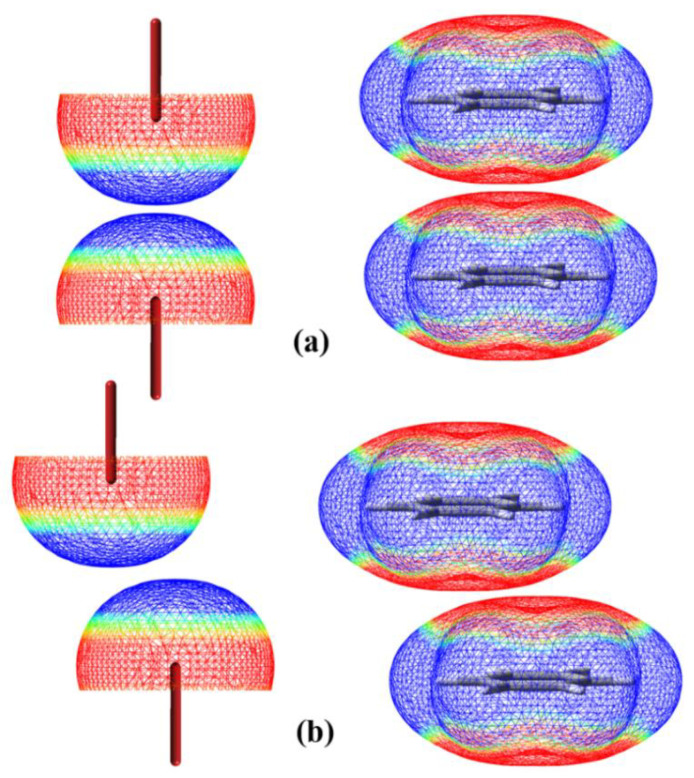
A schematic diagram of sandwich σ-hole···σ-hole stacking interaction versus sandwich π···π stacking interaction (**a**), and parallel-displaced σ-hole···σ-hole stacking interaction versus parallel-displaced π···π stacking interaction (**b**). The blue mesh represents the positive electrostatic potential and the red mesh represents the negative electrostatic potential.

**Figure 2 molecules-27-01252-f002:**
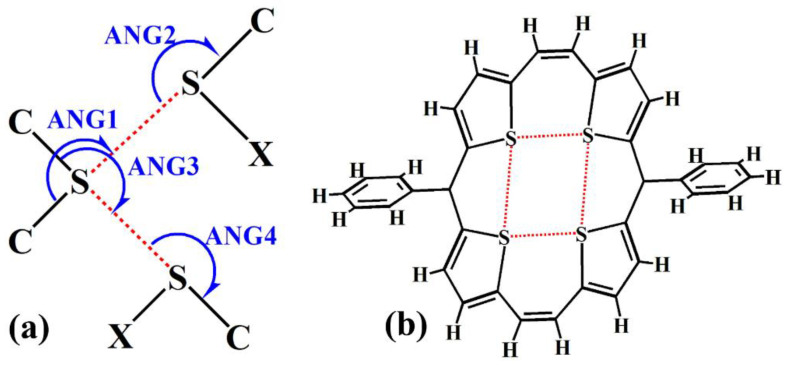
The schematic diagram of the intermolecular bifurcated σ-hole···σ-hole stacking interaction (**a**), and a typical example of the intramolecular bifurcated σ-hole···σ-hole stacking interactions in the crystal structure of 5,16-diphenyltetrathia[22]annulene (EJIGOU) (**b**).

**Figure 3 molecules-27-01252-f003:**
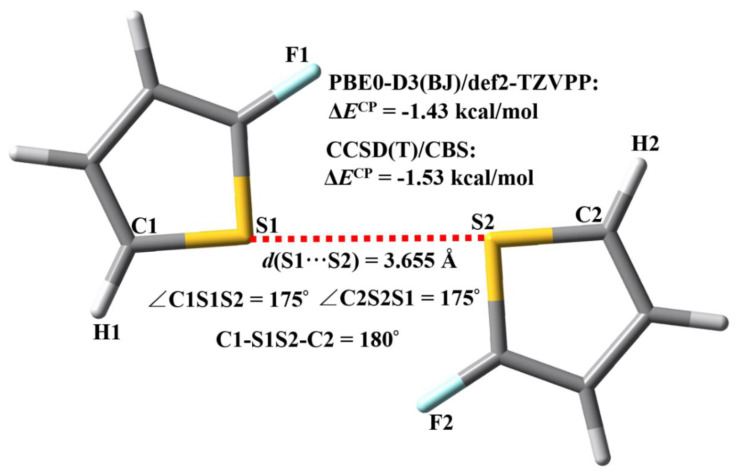
The PBE0-D3(BJ)/def2-TZVPP optimized geometries, PBE0-D3(BJ)/def2-TZVPP interaction energy and CCSD(T)/CBS interaction energy of the 2-fluorothiophene dimer.

**Figure 4 molecules-27-01252-f004:**
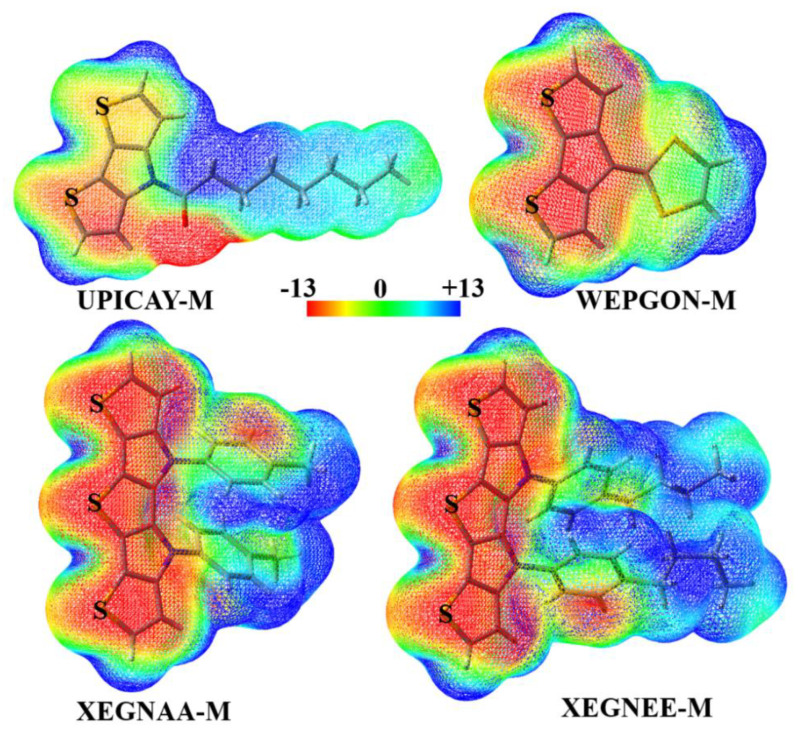
The molecular electrostatic potential maps of the monomers UPICAY-M, WEPGON-M, XEGNAA-M and XEGNEE-M. The color scale is in kcal/mol.

**Figure 5 molecules-27-01252-f005:**
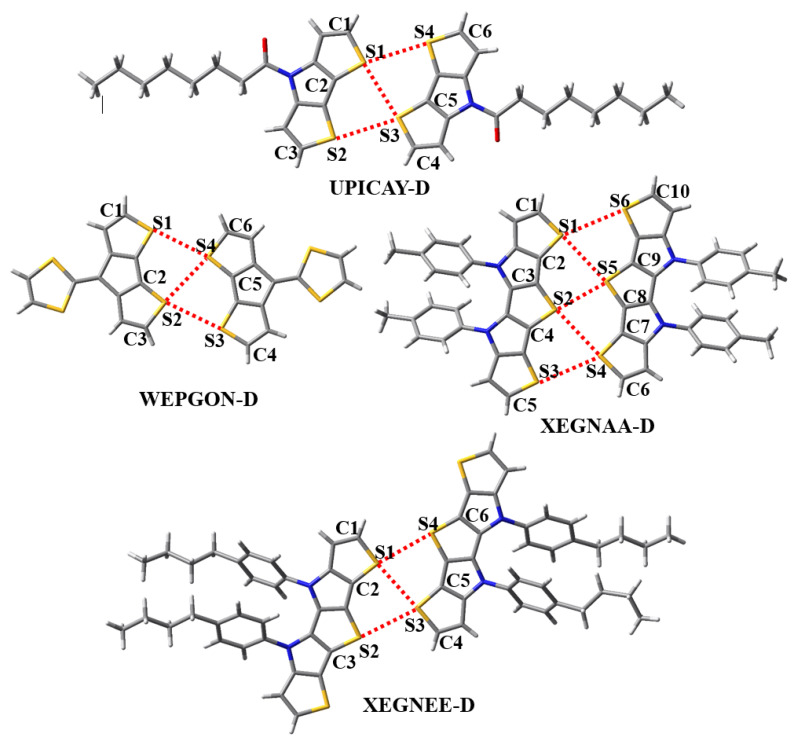
The bifurcated σ-hole···σ-hole stacking interactions (red dashed lines) in the dimers UPICAY-D, WEPGON-D, XEGNAA-D and XEGNEE-D. The N atom is shown in dark blue; O is shown in red and H is in white.

**Figure 6 molecules-27-01252-f006:**
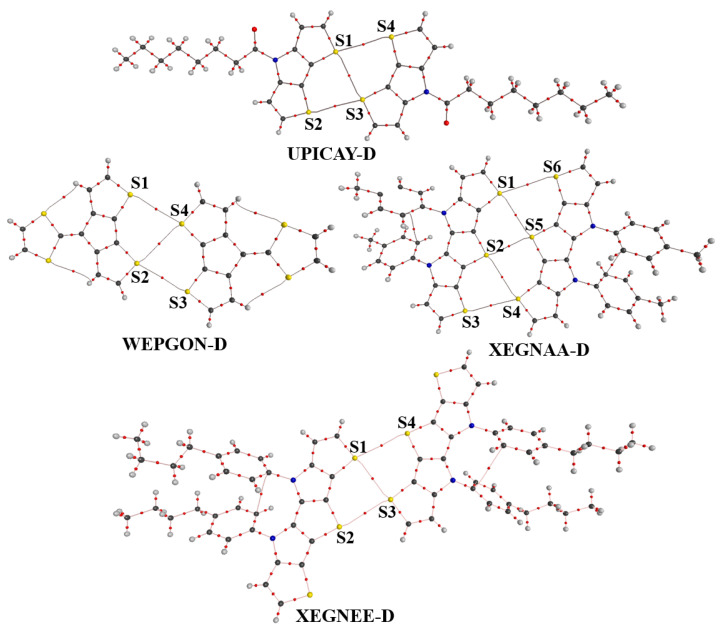
The bond paths and bond critical points (small red dots) of the dimers UPICAY-D, WEPGON-D, XEGNAA-D and XEGNEE-D.

**Figure 7 molecules-27-01252-f007:**
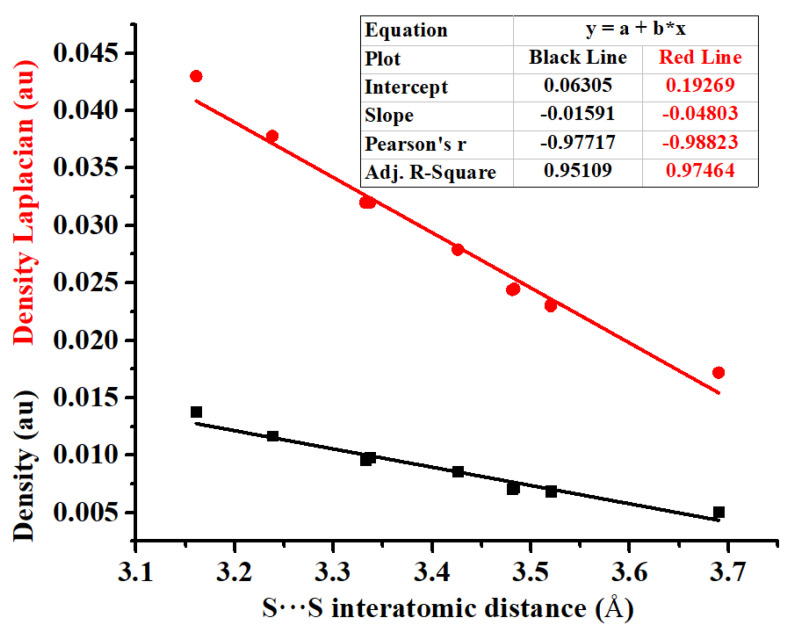
Correlation of the S···S interatomic distance with the electron density (black line) or electron density Laplacian (red line).

**Table 1 molecules-27-01252-t001:** The S···S interatomic distances (*d*_S···S_, Å), C–S···S angles (∠CSS, degree) and interaction energies ($E_{\mathrm{int}}^{\mathrm{CP}}$, kcal/mol) of the dimers studied.

Dimer	*D*	∠CSS	$E_{\mathrm{int}}^{\mathrm{CP}}$
UPICAY-D	*d*_S1···S3_ = 3.337	∠C1S1S3 = 173; ∠C4S3S1 = 173	−3.60
	*d*_S1···S4_ = 3.520	∠C2S1S4 = 166; ∠C6S4S1 = 176	
	*d*_S2···S3_ = 3.520	∠C3S2S3 = 176; ∠C5S3S2 = 166	
WEPGON-D	*d*_S1_···_S4_ = 3.426	∠C1S1S4 = 169; ∠C5S4S1 = 164	−3.16
	*d*_S2_···_S3_ = 3.426	∠C2S2S3 = 164; ∠C4S3S2 = 169	
	*d*_S2_···_S4_ = 3.520	∠C3S2S4 = 171; ∠C6S4S2 = 171	
XEGNAA-D	*d*_S1···S5_ = 3.481	∠C1S1S5 = 179; ∠C8S5S1 = 171	−4.73
	*d*_S1···S6_ = 3.690	∠C2S1S6 = 156; ∠C10S6S1 = 174	
	*d*_S2···S4_ = 3.481	∠C3S2S4 = 171; ∠C6S4S2 = 179	
	*d*_S2_···_S5_ = 3.161	∠C4S2S5 = 175; ∠C9S5S2 = 175	
	*d*_S3_···_S4_ = 3.690	∠C5S3S4 = 174; ∠C7S4S3 = 156	
XEGNEE-D	*d*_S1···S3_ = 3.238	∠C1S1S3 = 175; ∠C4S3S1 = 178	−3.40
	*d*_S1_···_S4_ = 3.332	∠C2S1S4 = 171; ∠C6S4S1 = 179	
	*d*_S2_···_S3_ = 3.483	∠C3S2S3 = 178; ∠C5S3S2 = 166	

**Table 2 molecules-27-01252-t002:** Energy decomposition of the SCS-SAPT0/aug-cc-pVDZ total interaction energies ($E_{\mathrm{int}}^{\mathrm{SCS}-\mathrm{SAPT}0}$) for the dimers studied. The PBE0-D3(BJ)/def2-TZVPP total interaction energies ($E_{\mathrm{int}}^{\mathrm{CP}}$) are listed for comparison. All energy values are given in kcal/mol.

	UPICAY-D	WEPGON-D	XEGNAA-D	XEGNEE-D
$E_{\mathrm{int}}^{\mathrm{CP}}$	−3.60	−3.16	−4.73	−3.40
$E_{\mathrm{int}}^{\mathrm{SCS}-\mathrm{SAPT}0}$	−3.58	−3.25	−4.83	−3.66
*E* _elst_	−3.82	−3.86	−5.48	−4.86
*E* _exch_	9.15	9.74	15.54	12.66
*E* _ind_	−1.41	−1.41	−2.41	−2.03
*E* _scs-disp_	−7.50	−7.73	−12.49	−9.43
*E*_elst_% ^1^	30%	30%	27%	30%
*E*_ind_% ^1^	11%	11%	12%	12%
*E*_scs-disp_ ^1^	59%	59%	61%	58%

^1^ Contribution to the total attractive interactions.

**Table 3 molecules-27-01252-t003:** The electron densities (*ρ*), electron density Laplacians (▽^2^*ρ*), eigenvalues of the Hessian matrix (*λ*_1_, *λ*_2_, *λ*_3_) and ellipticities (*ε*) at the bond critical points of the S···S contacts in the dimers studied. Atomic units are used for these quantities.

Dimer	Contact	*ρ* _b_	▽^2^*ρ*_b_	*λ* _1_	*λ* _2_	*λ* _3_	*Ε*
UPICAY-D	S1···S3	0.0098	0.0320	−0.0071	−0.0056	0.0446	0.2688
	S1···S4	0.0068	0.0230	−0.0046	−0.0035	0.0311	0.2983
	S2···S3	0.0068	0.0230	−0.0046	−0.0035	0.0311	0.2983
WEPGON-D	S1···S4	0.0086	0.0279	−0.0063	−0.0049	0.0391	0.2901
	S2···S3	0.0086	0.0279	−0.0063	−0.0049	0.0391	0.2901
	S2···S4	0.0069	0.0231	−0.0046	−0.0036	0.0313	0.2613
XEGNAA-D	S1···S5	0.0071	0.0244	−0.0048	−0.0036	0.0328	0.3061
	S1···S6	0.0051	0.0172	−0.0033	−0.0025	0.0230	0.3233
	S2···S4	0.0071	0.0244	−0.0048	−0.0036	0.0328	0.3061
	S2···S5	0.0138	0.0430	−0.0109	−0.0081	0.0620	0.3470
	S3···S4	0.0051	0.0172	−0.0033	−0.0025	0.0230	0.3233
XEGNEE-D	S1···S3	0.0117	0.0378	−0.0086	−0.0069	0.0533	0.2508
	S1···S4	0.0096	0.0320	−0.0069	−0.0052	0.0441	0.3263
	S2···S3	0.0072	0.0245	−0.0049	−0.0036	0.0330	0.3422

## Data Availability

Data is contained within the article.

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
