# Peer review of "The Bifurcated σ-Hole···σ-Hole Stacking Interactions"

_molecules, 2022, doi:10.3390/molecules27041252_

Round 1
Reviewer 1 Report
In this work, the authors employed computational chemistry method to investigate the bifurcated sigma-hole ... sigma-hole interaction between sulfur atoms. The authors did a search in the Cambridge Structural Database with a set of reasonable structural parameters and identified a set of materials that potentially feature the sigma-hole ... sigma-hole interaction. They then performed DFT and SAPT calculations to investigate the interaction, focusing on interaction energy, stacking structure, and bond analysis. Overall, the calculation methods are soundly chosen, and the conclusions are supported by the computational results. The paper may be accepted for publication with the following minor corrections.
1. I will ask the authors to provide color legend in Figure 1.
2. In line 80, I would ask the authors to remove "urgent". The research is certainly of interest. However, it is overselling to say it is of urgence.
3. I wonder whether the authors invented the term of "monofurcate". "Furcate" means branching into several channels. "Mono" means no branching. The term of "monofurcate" seems odd. It may be better to change "monofurcate" to "nonfurcate".
4. In line 113, "in the systems you are researching". This is too colloquial. It may be deleted.
5. In line 288, "in consistent with" shall be "in consistence with".
Author Response
In this work, the authors employed computational chemistry method to investigate the bifurcated sigma-hole ... sigma-hole interaction between sulfur atoms. The authors did a search in the Cambridge Structural Database with a set of reasonable structural parameters and identified a set of materials that potentially feature the sigma-hole ... sigma-hole interaction. They then performed DFT and SAPT calculations to investigate the interaction, focusing on interaction energy, stacking structure, and bond analysis. Overall, the calculation methods are soundly chosen, and the conclusions are supported by the computational results. The paper may be accepted for publication with the following minor corrections.
- I will ask the authors to provide color legend in Figure 1.
Reply: Figure 1 is a schematic diagram of the sandwich σ-hole···σ-hole stacking interaction versus sandwich π···π stacking interaction, and parallel-displaced σ-hole···σ-hole stacking interaction versus parallel-displaced π···π stacking interaction. In order to avoid the ambiguity, we have added “a schematic diagram of” before the Figure 1 caption.
- In line 80, I would ask the authors to remove "urgent". The research is certainly of interest. However, it is overselling to say it is of urgence.
Reply: We thank the reviewer for this suggestion and have deleted the word “urgent”.
- I wonder whether the authors invented the term of "monofurcate". "Furcate" means branching into several channels. "Mono" means no branching. The term of "monofurcate" seems odd. It may be better to change "monofurcate" to "nonfurcate".
Reply: The term “monofurcated” has appeared many times in the literature (Cryst. Growth Des. 2018, 18, 7579–7589; Cryst. Growth Des. 2021, 21, 5515–5520; Cryst. Growth Des. 2022, 22, 148–158).
- In line 113, "in the systems you are researching". This is too colloquial. It may be deleted.
Reply: We thank the reviewer for this suggestion and have deleted it.
- In line 288, "in consistent with" shall be "in consistence with".
Reply: We thank the reviewer for this suggestion and have changed “in consistent with” into “in consistence with”.
Reviewer 2 Report
Referee report on the manuscript entitled: „The Bifurcated σ-Hole···σ-Hole Stacking Interactions” written by Yu Zhang and Weizhou Wang.
This is a theoretical study of the σ-Hole···σ-Hole Stacking Interactions in four structures taken from the Cambridge Crystallographic Data Centre (CCDC). The Reviewer found the study interesting and worth publishing, but before it will be ready for publication in the Molecules journal, the Authors should clarify some issues listed below:
Lines 82 - 108 of the Introduction – the Reviewer suggests to move the part of the text to the Materials and Methods section.
The Authors should clearly define the main aim of their study. They should clarify as well the novelty and importance of their research.
Line 118 – the repetition of the word “the”
Was the Boys-Bernardi method applied during the geometry optimization or a posteriori ? Please clarify in the text of the manuscript.
Line 154 – there is written that the geometries were extracted from the X-ray data – it is not clear for me if the Authors performed the geometry optimization for them. In addition, which program did the Authors use for the electrostatic potential maps generation ?
Line 245 – the Authors should mention in the Materials and Methods section about HF/def2-TZVPP calculations to complete the computational protocol and introduce all used levels of theory.
According to the Reviewer’s opinion it would be good to enlarge the description of the SAPT and AIM methods application in the study. In the Materials and Methods section the Authors should prepare a larger introduction of the methods with technical details used in the study. It would be of interest for potential Readers of the manuscript.
Author Response
This is a theoretical study of the σ-Hole···σ-Hole Stacking Interactions in four structures taken from the Cambridge Crystallographic Data Centre (CCDC). The Reviewer found the study interesting and worth publishing, but before it will be ready for publication in the Molecules journal, the Authors should clarify some issues listed below:
Lines 82 - 108 of the Introduction – the Reviewer suggests to move the part of the text to the Materials and Methods section.
Reply: We have moved lines 82-108 to the Materials and Methods section.
The Authors should clearly define the main aim of their study. They should clarify as well the novelty and importance of their research.
Reply: The study of the organic optical and electronic materials is very hot in recent years. The thiophene-based organic molecules and polymers have been found to play key roles in the development of novel organic optical and electronic materials. Therefore, it is important to study the noncovalent interactions involving the S atom. We have emphasized the novelty and importance of this research in lines 74-81.
Line 118 – the repetition of the word “the”
Reply: We have deleted the word “the”.
Was the Boys-Bernardi method applied during the geometry optimization or a posteriori ? Please clarify in the text of the manuscript.
Reply: We have added the following sentence in the Materials and Methods section:
“The geometries of these monomers and dimers were all extracted from the crystal structures and without further optimizations.”
Line 154 – there is written that the geometries were extracted from the X-ray data – it is not clear for me if the Authors performed the geometry optimization for them. In addition, which program did the Authors use for the electrostatic potential maps generation ?
Reply: We have added the following two sentences in the Materials and Methods section:
“The geometries of these monomers and dimers were all extracted from the crystal structures and without further optimizations.”
“The molecular electrostatic potentials and interaction energies were calculated with the GAUSSIAN 09 program package [36].”
Line 245 – the Authors should mention in the Materials and Methods section about HF/def2-TZVPP calculations to complete the computational protocol and introduce all used levels of theory.
Reply: We thank the reviewer for this suggestion and have added the following sentence in the Materials and Methods section:
“For comparison, the CP-corrected HF/def2-TZVPP interaction energies of the dimers were also calculated.”
According to the Reviewer’s opinion it would be good to enlarge the description of the SAPT and AIM methods application in the study. In the Materials and Methods section the Authors should prepare a larger introduction of the methods with technical details used in the study. It would be of interest for potential Readers of the manuscript.
Reply: We thank the reviewers for very helpful suggestions. We have added the following sentences to enlarge the description of the SAPT and AIM methods. We also wish the reviewer can understand that, due to the article duplicate check, it is unsuitable to present much more details of the two methods.
“Sherrill and coworkers developed various SAPT methods for the energy component analyses of noncovalent interactions [31-33]. The computations of the high-order SAPT methods are very expensive and cannot be employed for the study of the large systems. The low-order SAPT0 method neglects the intramonomer electron correlation and can treat the systems with hundreds of atoms. It has been reported that the combination of the SCS-SAPT0 method with the basis set aug-cc-pVDZ has a very high accuracy in analyzing the noncovalent interactions in the large π-stacked complexes [34].”
“The AIM theory defines chemical bond and structure of a chemical system based on the topological analysis of the electron density and its Laplacian. It has gradually become a very valuable tool for addressing possible questions regarding the noncovalent interactions.”
Reviewer 3 Report
The manuscript “The Bifurcated sigma-Hole…sigma-Hole Stacking Interactions” by Zhang and Wang describes bifurcated noncovalent interactions that result from sigma holes on S atoms formed by C-S bonds. The manuscript is very nicely, clearly and concisely written, and it sheds light on the vast field of noncovalent interactions. I suggest its publication after the authors consider the following suggestions.
1. Page 7, line 207. It is stated that the sigma-hole…sigma-hole interaction is slightly attractive or even repulsive. In order to be more precise, it should be emphasized that the S…S distance determines whether the interaction is attractive or repulsive. Otherwise, one would think that both attraction and repulsion refer to the crystal structure.
2. Page 7, line 218: “Normally the absolute value of the interaction energy of XEGNAA-D should be larger than 5.00 kcal/mol.” Could the authors elaborate the origin of 5.00 kcal/mol in this statement?
Author Response
The manuscript “The Bifurcated sigma-Hole…sigma-Hole Stacking Interactions” by Zhang and Wang describes bifurcated noncovalent interactions that result from sigma holes on S atoms formed by C-S bonds. The manuscript is very nicely, clearly and concisely written, and it sheds light on the vast field of noncovalent interactions. I suggest its publication after the authors consider the following suggestions.
- Page 7, line 207. It is stated that the sigma-hole…sigma-hole interaction is slightly attractive or even repulsive. In order to be more precise, it should be emphasized that the S…S distance determines whether the interaction is attractive or repulsive. Otherwise, one would think that both attraction and repulsion refer to the crystal structure.
Reply: We thank the reviewer for the helpful suggestion. We have changed the sentence into “The S···S interatomic distance will determine whether the monofurcated σ-hole(S)···σ-hole(S) stacking interaction is attractive or repulsive.”
- Page 7, line 218: “Normally the absolute value of the interaction energy of XEGNAA-D should be larger than 5.00 kcal/mol.” Could the authors elaborate the origin of 5.00 kcal/mol in this statement?
Reply: We have added two sentences to further explain the origin of 5.00 kcal/mol:
“This means that the absolute value of the interaction energy of one monofurcated σ-hole···σ-hole stacking interaction is larger than 1.00 kcal/mol. The interaction energies of the dimer XEGNAA-D is -4.73 kcal/mol. Normally the absolute value of the interaction energy of XEGNAA-D should be larger than 5.00 kcal/mol because it has five monofurcated σ-hole···σ-hole stacking interactions.”
Reviewer 4 Report
This article probes the possibility of attractive interactions between the positive regions associated with sigma holes on a pair of chalcogen atoms. Due to the near linearities of the intermolecular geometries, their identification as chalcogen bonds is precluded, so the authors dissect the interactions to better understand their underlying nature. The level of theory is quite high, so data can be considered definitive. The means of analysis are appropriate, using AIM to identify interatomic nocovalent bonding propensities, and SAPT to analyze the forces involved. The analogy to pi-pi stacking presented in the Introduction proves apt, as the interactions are dominated by dispersion with only modest boosts from electrostatics and induction. The text is written efficiently, and ought to be accessible to most people interested in this topic. Tables and figures are designed with care and highlight the salient issues. The work represents a step forward in this area, so publication is recommended. There are only several minor issues which require some attention.
First, does the optimized geometry of the pilot system presented in Fig 3 represent a true minimum, with all positive vibrational frequencies?
What is the origin or definition of the red lines in Fig 5; are they merely to indicate S--S axes, or do they convey something more definitive?
It is unfortunate that the authors have chosen to call these interactions by the name of sigma hole-sigma hole. This designation implies some sort of electrostatic attraction between these regions which is not the case and will cause confusion to readers. The slightly negative values of the full intermolecular ES component of SAPT arises despite a fundamental repulsion between these holes, not because of them. And the ES term is much smaller than dispersion in any case. After all, the analogous interactions in Fig 1 are not called lump-lump, negative-negative, or anti-electrostatic, nor is any reference made to their facing negative regions. They are typically called pi-pi, or sandwich, or stacked, terms which reference their geometry or the segments of their electronic structure. It would be less misleading to simply refer to the interactions described here as simply dispersive.
Author Response
This article probes the possibility of attractive interactions between the positive regions associated with sigma holes on a pair of chalcogen atoms. Due to the near linearities of the intermolecular geometries, their identification as chalcogen bonds is precluded, so the authors dissect the interactions to better understand their underlying nature. The level of theory is quite high, so data can be considered definitive. The means of analysis are appropriate, using AIM to identify interatomic noncovalent bonding propensities, and SAPT to analyze the forces involved. The analogy to pi-pi stacking presented in the Introduction proves apt, as the interactions are dominated by dispersion with only modest boosts from electrostatics and induction. The text is written efficiently, and ought to be accessible to most people interested in this topic. Tables and figures are designed with care and highlight the salient issues. The work represents a step forward in this area, so publication is recommended. There are only several minor issues which require some attention.
First, does the optimized geometry of the pilot system presented in Fig 3 represent a true minimum, with all positive vibrational frequencies?
Reply: As written in lines 153 and 154, the geometries of these monomers and dimers were all extracted from the crystal structures and without further optimizations. These organic optical and electronic materials are in solid state.
What is the origin or definition of the red lines in Fig 5; are they merely to indicate S--S axes, or do they convey something more definitive?
Reply: As stated in the caption of Fig. 5, the red dashed lines represent the bifurcated σ-hole···σ-hole stacking interactions in the dimers. The definition of the bifurcated σ-hole···σ-hole stacking interaction was discussed in lines 82-109.
It is unfortunate that the authors have chosen to call these interactions by the name of sigma hole-sigma hole. This designation implies some sort of electrostatic attraction between these regions which is not the case and will cause confusion to readers. The slightly negative values of the full intermolecular ES component of SAPT arises despite a fundamental repulsion between these holes, not because of them. And the ES term is much smaller than dispersion in any case. After all, the analogous interactions in Fig 1 are not called lump-lump, negative-negative, or anti-electrostatic, nor is any reference made to their facing negative regions. They are typically called pi-pi, or sandwich, or stacked, terms which reference their geometry or the segments of their electronic structure. It would be less misleading to simply refer to the interactions described here as simply dispersive.
Reply: We thank the reviewer for helpful comments. It is generally accepted that the noncovalent interaction arises from a precise balance between electrostatic energy, exchange repulsion energy, induction energy and dispersion energy. Hence, we used the word “dispersion-dominated” in this study. We also emphasized the important role of the electrostatic energy.
The σ-hole···σ-hole stacking interactions are electrostatically attractive not electrostatically repulsive as expected. It is the short-range charge penetration that causes the electrostatically attractive interaction between two σ-hole. We explained this unexpected phenomenon in lines 270-275.